# Identification of Endothelial Cell Protein C Receptor by Urinary Proteomics as Novel Prognostic Marker in Non-Recovery Kidney Injury

**DOI:** 10.3390/ijms25052783

**Published:** 2024-02-28

**Authors:** Chih-Hsiang Chang, Cheng-Chia Lee, Yung-Chang Chen, Pei-Chun Fan, Pao-Hsien Chu, Lichieh Julie Chu, Jau-Song Yu, Hsiao-Wei Chen, Chih-Wei Yang, Yi-Ting Chen

**Affiliations:** 1Kidney Research Center, Department of Nephrology, Chang Gung Memorial Hospital, Linkou Branch, Taoyuan 333, Taiwan; sunchang@cgmh.org.tw (C.-H.C.); chia7181@gmail.com (C.-C.L.); cyc2356@gmail.com (Y.-C.C.); franwis1023@gmail.com (P.-C.F.); cwyang00@gmail.com (C.-W.Y.); 2Graduate Institute of Clinical Medicine Science, College of Medicine, Chang Gung University, Taoyuan 333, Taiwan; 3Department of Cardiology, Chang Gung Memorial Hospital, Linkou Branch, Taoyuan 333, Taiwan; taipei.chu@gmail.com; 4Molecular Medicine Research Center, Chang Gung University, Guishan, Taoyuan 333, Taiwan; julie.chu@mail.cgu.edu.tw (L.J.C.); yusong@mail.cgu.edu.tw (J.-S.Y.); lara0714@mail.cgu.edu.tw (H.-W.C.); 5Graduate Institute of Biomedical Sciences, College of Medicine, Chang Gung University, Taoyuan 333, Taiwan; 6Department of Biomedical Sciences, College of Medicine, Chang Gung University, Taoyuan 333, Taiwan

**Keywords:** acute kidney injury, acute decompensated heart failure, endothelial cell protein C receptor (EPCR), fibrosis

## Abstract

Acute kidney injury is a common and complex complication that has high morality and the risk for chronic kidney disease among survivors. The accuracy of current AKI biomarkers can be affected by water retention and diuretics. Therefore, we aimed to identify a urinary non-recovery marker of acute kidney injury in patients with acute decompensated heart failure. We used the isobaric tag for relative and absolute quantification technology to find a relevant marker protein that could divide patients into control, acute kidney injury with recovery, and acute kidney injury without recovery groups. An enzyme-linked immunosorbent assay of the endothelial cell protein C receptor (EPCR) was used to verify the results. We found that the EPCR was a usable marker for non-recovery renal failure in our setting with the area under the receiver operating characteristics 0.776 ± 0.065; 95%CI: 0.648–0.905, (*p* < 0.001). Further validation is needed to explore this possibility in different situations.

## 1. Introduction

Acute kidney injury (AKI) is a common and severe complication in critically ill patients. Despite the progression of medical care techniques, the mortality rate remains high and patients who survive AKI usually experience chronic kidney disease (CKD), hypertension, and even progress to EKSD [1]. The costs and medical expenditures of AKI are high, as is the care burden on families, society, and the health care system. In the past, the definition of AKI differed between studies. To standardize and increase the sensitivity for early identification of AKI, the Kidney Disease: Improving Global Outcomes (KDIGO) group defined AKI as an increase in serum Cr to 1.5 times the baseline value within 7 d or a urine output less than 0.5 mL/kg/h for 6 h and a serum Cr level increase exceeding 0.3 mg/dL within 48 h [2,3]. However, serum Cr is a delayed functional marker that is elevated 48–72 h after kidney injury [4]. This is expected to delay management and intervention.

### 1.1. Clinical Needs for Novel AKI Biomarkers in Heart Failure

In the past decade, researchers have sought to develop biomarkers for earlier detection of AKI. Many promising AKI biomarkers have been reported, such as neutrophil gelatinase-associated lipocalin (NGAL), kidney injury molecule one (KIM-1), liver type fatty acid-binding protein (L-FABP), and interleukin-18 (IL-18), and calculation of [IGFBP7] × [TIMP-2]. However, there is some ambiguity about using these markers in patients admitted for acute decompensated heart failure (ADHF) with worsening renal function (WSF) [5]. Among these patients, it would be particularly helpful to identify those who are progressing and appear refractive to adjustments, such as those made using inotropic agents, medications, and fluid. Therefore, we herein aim to use a cutting-edge proteomic method to accurately identify a marker for AKI progression without recovery under adjustment (non-recovery AKI) in patients with heart failure. 

### 1.2. The Rationale for Discovering an AKI Biomarker from the Urine Proteome

The apical side of the renal tubule is in direct contact with the urine, which is the primary fluid of the urological system and therefore may contain information associated with kidney function status and disease progression. Under ischemia, the renal tubule in the outer medulla becomes damaged due to hypoxia. In an early stage of this process, the renal cells will try to adapt to the stress. Once hypoxia progresses, however, the cells will undergo apoptosis or necrosis. If the renal damage becomes severe, the renal tubules are obstructed; this will worsen renal function and may alter the proteomic profile of urine [6]. Moreover, urine screening is a non-invasive and high-compliance test. Recent advances in modern proteomics techniques can enable researchers to discover and verify novel biomarkers by assessing the initial change of biological information early in disease progression [7,8]. In the present study, we decided to use a high-resolution chemical labeling proteomic platform called isobaric tag for relative and absolute quantification (iTRAQ) [9] to identify a protein marker for non-recovery AKI.

### 1.3. Biomarker Discovery by Shotgun Proteomics

Coupling large-scale sequencing projects with the amino acid sequence information gleaned from tandem mass spectrometry (MS/MS) has made it much easier for researchers to analyze the complicated composition of proteins in body fluids. The limits of a “shotgun” approach, in which the protein mixture is proteolytically digested before separation, can be expanded by separating the resulting peptide mixture before MS/MS analysis. Multidimensional liquid chromatography (LC) can be directly interfaced with the mass spectrometer to enable the automated collection of tremendous quantities of data for protein identification and quantification. While no single technique addresses all proteomic challenges, the shotgun approach shows advantages over gel-based processes in its speed, sensitivity, analytic scope, and dynamic range for novel protein identification and quantification [10,11]. 

Therefore, this study aims to utilize an advanced proteomic approach, iTRAQ, to 20 January 2024 identify a novel protein marker for non-recovery AKI in patients with ADHF.

## 2. Results

### 2.1. AKI-Associated Changes of the Urine Proteome

In the biomarker discovery phase, we set out to discover potential biomarker candidates. Equal amounts of urinary proteins from 15, 9, and 6 individuals (Table 1) were digested by trypsin and pooled as three age-matched subgroups: control, AKI with the control, and AKI with recovery and non-recovery AKI subgroups were labeled with 4-plexed iTRAQ reagents bearing tags 114, 115, and 116, respectively. Mixed iTRAQ-labeled peptides of the three subgroups were extensively analyzed using 2DLC iTRAQ. Protein identification and quantification results obtained from the iTRAQ experiments are detailed in Table 1. The peak area ratios for protein expression levels were determined by the iTRAQ reporter ions and used to select biomarker candidates. The normal group was used as a control to evaluate the fold changes of protein concentrations between AKI groups with recovery and non-recovery. A total of 2384 urine proteins were identified in the iTRAQ experiment, 2308 of which yielded at least one quantifiable peptide; of them, 1177 proteins did not show any significant concentration difference between the AKI groups and control groups. The proteins displaying a fold difference ≥2 or ≤0.5 between any AKI subgroup and the control subgroup were measured as the 115/114 and 116/114 ratios, respectively, and were defined as increased or decreased proteins, respectively. We identified 189 increased and 109 decreased proteins in two AKI groups. The identified up- and down-regulated proteins are listed in Appendix A. The protein selection and verification for AKI non-recovery was performed through the review of the top 20 proteins with either a higher 116/115 ratio or a higher 116/114 ratio. The proteins were evaluated based on prior research articles, pathophysiological relevance, and availability of commercialized ELISA kits. After several protein verifications, EPCR was selected due to its potential functional relevance to non-recovery AKI. The differential expressions of these increased or decreased proteins are shown as the heat map in Figure 1 to provide the global changes of the two AKI groups.

### 2.2. Pathway Analysis of Dysregulated Proteins

Our MetaCore analysis identified two potentially important pathways, one associated with protein folding (Figure 2) and one associated with complement and inflammation (Figure 3A,B).

### 2.3. Protein Folding, Maturation, and POMC Processing Pathways

Acute kidney injury results in the accumulation of unfolded and misfolded proteins in the endoplasmic reticulum (ER), leading to the unfolded protein response (UPR) or ER stress [12]. The UPR can be successfully solved by inducing chaperonins to increase the ER folding capacity [13], or degrading these abnormal proteins via ER-associated degradation [14]. Once the damage is not successfully addressed, the apoptotic pathway is initiated. In this case, IRE-1 would stimulate the JNK pathway, which regulates cell death/survival and downstream proinflammatory cytokines to promote the inflammation [15]. The precursor protein, pro-opiomelanocortin (POMC), gives rise to melanocortin peptides (ACTH, α-MSH, β-MSH, and γ-MSH) that are important for dealing with ER stress during AKI [16]. ACTH ameliorates the severity of kidney injury in animal models [17]. The result of EKGG pathway analysis also showed that the levels of MSH, ACTH, and PMOC are higher in patients without AKI than in those with AKI. Melanocortin peptides protect the kidney through at least two mechanisms: (1) stimulating the production of corticosteroids and (2) activating melanocortin receptors (MCRs) expressed by diverse kidney parenchymal cells. Since this did not differ between the recovery and non-recovery groups, we speculate that ER stress and protein misfolding are an initial response to AKI but not a crucial parameter for non-recovery AKI.

### 2.4. Immune Responses Via the Alternative Complement Pathway and Lectin-Induced Pathway

Lectin pathway (LP)-mediated complement activation is part of the innate immune system and has been implicated in various pathophysiologies, including IgA nephropathy [18,19], Henoch–Schonlein purpura nephritis [20], diabetic nephropathy [21], and ischemia–reperfusion injury [22]. P begins with the attachment of specific LP-pattern recognition elements. Two kinds of LP recognition elements are recognized: mannan-binding lectin (MBL) and collectin-11 (CL-11). Both can attach to carbohydrate structures, and ficolins that connect to acetylated ligand patterns found on pathogens and damaged cells [23]. The attachment of LP-recognition complexes to their specific ligands activates MBL-associated serine protease zymogens into their functional enzyme form, propelling LP-driven complement (C) activation. In this progression, MASP-2 plays a crucial role in leading the LP-C3 convertase, C4b2a. MASP-2 is unique as it is the sole MASP capable of splitting both complement C2 and C4, a process that results in the conversion of C3 to C3a. Our KEGG results revealed that in the alternative pathway, C3 and C3a were elevated in the non-recovery group, as were C3c and C3dg. Moreover, in the lectin pathway, MLB2 and L-ficolin also contributed to the C3b and C5 convertase [24]. The above changes result in C9, lysis of target cells, and damage to vessels and tissues.

### 2.5. Verification of EPCR by ELISA

A total of 51 patients with AKI and ADHF who were not part of the discovery group were enrolled and divided into recovery and non-recovery groups, and ELISA was used to test the ability of EPCR to distinguish between these groups. The characteristic and demographic data for these patients are presented in Table 2. The patients in both groups were similar in age, sex, diabetes, and medication. The APACHE and SOFA scores were also similar. The baseline Cr was not significantly different. The AUROC of urine EPCR/Cr was 0.776 ± 0.065 (95%CI: 0.648–0.905), as shown in Figure 4. And the multi-regression model showed the EPCR/Cr remained significant even adjusted for age, sex, DM, shock, and medications (Appendix A). This argues for the use of EPCR as a marker for non-recovery AKI. 

## 3. Discussion

### 3.1. Advances in Urinary Proteome Identification

Non-invasive diagnosis and prognosis of kidney diseases are still challenges in clinical nephrology. The definition of biomarkers based on proteomic analysis, especially of urine, has advanced recently and may provide new tools to solve these challenges. The method applied in this article highlights a most promising technological approach for deciphering the human proteome and applying the knowledge in clinical nephrology, with an emphasis on the urinary proteome. Continued progress in human proteomic analysis in health and disease will likely depend on the standardization of data and the availability of suitable bioinformatics and software solutions, rather than new technological advances per se. The current literature indicates that although a thorough hands-on investigation of the entire urinary proteome is still challenging, the application of algorithms and/or artificial intelligence is likely to be the future in proteomic research. It is also predicted that proteomics will play an essential role in clinical nephrology [25].

### 3.2. Difficulties in Urinary Biomarker Verification

Untargeted proteomics analysis can generate a panel of candidate disease biomarkers, but further verification through targeted absolute quantification is necessary for clinical reasoning and applications. It is expensive and challenging for researchers to verify a list of candidate biomarkers, especially in the complex setting of clinical practice [26]. Antibody-based immunoassays, particularly ELISA, are currently the most common techniques for biomarker validation [27]. ELISA is highly sensitive and specific, but it is limited by the availability of antibodies and the expense of assay development for new target analytes [28,29]. In addition, ELISA is relatively time-consuming and costly when a large number of samples must be run. Using ELISA to verify target proteins suggested by iTRAQ is also tricky because iTRAQ shows the relative ratio of protein changes in different groups, while ELISA is used to detect protein concentrations. Therefore, it is important to perform repeat measurements and use a standard, such as the urine Cr level applied herein, to correct for biomarker concentrations.

### 3.3. Protein Function of EPCR and Relationship with Non-Recovery of AKI

The thrombomodulin (TM)–protein C system is vital for maintaining healthy endothelial cells and thereby supporting the maintenance of regional microcirculation and preventing thrombosis [30]. The cleavage of protein C by thrombin on the cell surface, which requires TM as a cofactor, generates activated protein C (APC). APC acts as an essential modulator of coagulation and sepsis-associated inflammation by inactivating factors Va and VIIIa, thereby promoting fibrinolysis and inhibiting thrombosis. In addition, soluble thrombomodulin (sTM), independent of its ability to generate APC, can reduce ischemia–reperfusion injury [31]. In the IRI mouse model, sTM reportedly attenuated the rise in creatinine after reperfusion, improved microvascular erythrocyte flow, reduced microvascular endothelial leukocyte adhesion, and minimized endothelial dysfunction permeability [32]. Soluble TM was found to act via endothelial cell protein C receptor (EPCR) to have cellular effects, including anti-inflammatory activity, anti-apoptotic activity, leukocyte activation, and stabilization of barrier function [31,33]. The binding of thrombin to thrombomodulin on the endothelial surface promotes its anticoagulant properties via APC and the thrombin–thrombomodulin complex. These molecules with preserved cytoprotective properties maintain renal blood flow (RBF) and attenuate acute kidney injury (Figure 5). Therefore, the lower EPCR/Cr detected in the urine of non-recovery AKI patients might reflect a lower level of EPCR in the kidney and more vascular damage, which would prolong kidney injury. 

### 3.4. Limitations

First, we explored AKI biomarkers in severe ADHF patients; further analysis of patients with less severe HF or those without admission might result in different outcomes. Second, this study involved a relatively small number of cases and a racially homogeneous population; thus, a future study involving a larger, more diverse population is needed in the future. Third, we did not exam the function of EPCR in animal models of renal fibrosis, so further studies are warranted to validate these findings presented herein. 

## 4. Methods and Materials

### 4.1. Patient Enrollment

The study protocol was approved by the local Institutional Review Board [201601783B0]. Patients with acute decompensated heart failure (ADHF) admitted to the intensive care unit (ICU) were enrolled between December 2016 and January 2020. Patients who were receiving dialysis were CKD stage 5, were aged <18 years, had cancer, and/or reported prior organ transplantation was excluded. Patients were eligible for enrollment within 3 d after admission, and urine was collected on the same day after enrollment. Progressive heart failure signs or symptoms that need hospital admission are defined as ADHF. The first 30 patients were enrolled as the discovery group in the iTRAQ study, and then the 51 patients were enrolled as the verification group in the ELISA study. AKI was defined as that occurring within 7 d after admission and was defined under the KDIGO clinical practice guidelines for AKI, which confirm AKI under any of the following conditions: serum Cr level increase ≥0.3 mg/dL within 48 h, or ≥1.5 times increase in serum Cr level from baseline within 7 d. In addition, the severity of AKI was staged according to the KDIGO guideline. Non-recovery AKI was defined as serum Cr more than 1.5 times the baseline after 2 weeks. Patients who enrolled in the trial and did not show evidence of AKI within 7 d were put in a control group. The following data were collected prospectively: demographic characteristics, primary diagnosis, routine biochemistry test results, treatment for heart failure, and outcomes. 

### 4.2. Collection and Pretreatment of Urine Samples

The urine samples were collected using a protease inhibitor cocktail tablet (one tablet per 50 mL urine; Roche, Mannheim, Germany) and sodium azide (1 mM). The collected samples were centrifuged at 5000× *g* for 30 min at 4 °C to remove cells and debris, and the clarified supernatants were stored at −20 °C for further processing [10,11,34,35].

Urine proteins were enriched using a 10-kDa centrifugal filter (Millipore, Carrigtwohill, Ireland). Urine samples (12.5 mL) were centrifuged at 5000× *g* for 30 min at 4 °C in the filter tube, and then the tube was decanted and refilled with 12.5 mL of 20% acetonitrile/H_2_O and centrifuged again. This process was repeated twice, with pure water used for desalting. The amount of protein in each concentrated/desalted urine sample was measured using a BCA protein assay kit (Bio-Rad, Hercules, CA, USA), and the samples were lyophilized and stored at −80 °C for subsequent processing. 

### 4.3. iTRAQ Labeling and Fractionation of Labeled Peptides

Equal amounts of urinary proteins from each individual of the same clinical status were pooled for the following sample preparation, as done in previous works [10,35]. Urine proteins from 15 NT volunteers, 9 AKI with recovery patients, and 6 AKI non-recovery patients were separately pooled and digested with sequencing-grade modified trypsin (Promega, Madison, WI, USA) at 37 °C overnight. According to the manufacturer’s instructions, the desalted peptides were labeled with iTRAQ tags 114, 115, and 116 (Applied Biosystems, Foster City, CA, USA). The three iTRAQ-tagged peptide mixtures were then pooled and desalted using an Oasis HLB 96-well µElution plate (Waters, Wexford, Ireland). The samples were dried by vacuum centrifugation and stored at −20 °C for LC-ESI MS/MS analysis.

### 4.4. LC-ESI MS/MS Analysis by Orbitrap-Elite MS

The iTRAQ-labeled peptides (20 µg) were reconstituted in 50 µL of HPLC buffer A (30% acetonitrile/0.1% formic acid) and loaded onto a homemade column (Luna SCX 5 µm, 0.5 × 180 mm) at a flow rate of 5 µL/min for 30 min. Forty-four peptide fractions were obtained by elution with 0–100% HPLC buffer B (0.5 M ammonium chloride/30% acetonitrile/0.1% formic acid) using an online 2D-HPLC apparatus (Dionex Ultimate 3000, Thermo Fisher, San Jose, CA, USA). Each SCX fraction was diluted in-line and then trapped onto a reverse-phase column (Zorbax 300SB-C18, 0.3 × 5 mm; Agilent Technologies, Wilmington, DE, USA). The desalted peptides were then separated on a homemade column (HydroRP 2.5 µm, 75 μm I.D., 20 cm with a 15-μm tip) using a linear gradient of 3–28% HPLC buffer C (99.9% acetonitrile/0.1% formic acid) for 37 min, 28–50% buffer C for 12 min, 50–95% buffer C for 2 min, 95% buffer C for 5 min, and 3% buffer C for 9 min, at a flow rate of 0.3 μL/min. The LC apparatus was coupled with a 2D linear ion trap mass spectrometer (LTQ-Orbitrap ELITE; Thermo Fisher, San Jose, CA, USA) operated using the Xcalibur 2.2 software (Thermo Fisher, San Jose, CA, USA). The full-scan MS was performed in the Orbitrap over a range of 400 to 2000 Da and a resolution of 60,000 at *m*/*z* 400. Internal calibration was performed using the ion signals of [Si(CH_3_)_2_O]6H^+^ at *m*/*z* 445.120025, 462.146574, and 536.165365 as lock masses. The 12 data-dependent MS/MS scan events (6 CID, 6 HCD) were followed by one MS scan for the six most abundant precursor ions in the preview MS scan. The *m*/*z* values selected for MS/MS were dynamically excluded for 40 s with a relative mass window of 1.5 Da. The electrospray voltage was set to 1.8 kV and the capillary temperature was set to 220 °C. The MS and MS/MS automatic gain controls were set to 1000 ms (full scan), 150 ms (CID), and 300 ms (HCD), or 2 × 10^6^ ions (full scan), 5 × 10^3^ ions (CID), and 3 × 10^4^ ions (HCD) for maximum accumulated time or ions, respectively. The LC-MS/MS raw data from the iTRAQ experiment have been deposited in the Appendix A.

### 4.5. MS Data Processing, Database Searching for Protein Identification, Protein Quantification, and Target Protein Selection

The individual raw data files were processed using Proteome Discoverer (version 1.3.0.339, Thermo Fischer Scientific, Waltham, MA, USA), then searched using Mascot 2.2 search engines and the Swissprot2010 database. The precursor mass tolerance was 10 ppm, and product ions were searched at 0.5-Da tolerances. Peptides were generated from tryptic digestion with up to two missed cleavages. The static modification was set as carbamidomethylation of cysteines, and the dynamic modifications were set to N-terminal glutamine deamidation, methionine oxidation, N-terminal acetylation, and N-terminal and lysine iTRAQ4plex labeling. Peptide spectral matches (PSM) were validated using a percolator based on q-values at a 1% fixed false discovery rate (FDR). Data analyses for the heat map were performed using the Partek Genomic Suite 6.6 software. To find the target protein to be verified by ELISA, we tested the protein by ELISA according to the possible protein function in biology and the ratio of target protein between the AKI recovery and non-recovery groups. We found that Cadherin-1, IL-18, Protein S100 A7, LPS, Uromodulin, and TNF-family were already reported in the literature. Verifying Semenogelin-1, Semenogelin-2, and apolipoprotein-C showed no association with the AKI non-recovery. Therefore, we chose the endothelial protein C receptor as a marker to detect AKI non-recovery.

### 4.6. ELISA for Quantification of Urinary EPCR Levels

A commercial ELISA kit (DY2245, Human EPCR Duoset ELISA kit; R&D Minneapolis, USA) was used per the manufacturer’s instructions. The result of urine EPCR level was normalized with urine Cr to exclude the confounding factor of urine volume. First, the captured antibody was diluted to a working concentration of 0.8 μg/mL in diluent reagent (1% BSA in PBS, pH 7.2–7.4, 0.2-μm filtered). Then, 100 μL per well of antibody solution was added and the sealed plate was incubated overnight at room temperature. After the antibody was coated completely, the plate was washed three times with washing buffer (0.05% Tween^®^ 20 in PBS). Diluent reagent (300 μL/well) was added, and the plate was incubated at room temperature for 1 h and washed three times. For quantification, a seven-point standard curve was constructed using 2-fold serial dilutions (maximum: 20 ng/mL).

Each raw urine sample was prepared as a 10-fold dilution using H_2_O, and 100 μL of diluted urine sample or standard was added into each well. The well was then sealed, incubated for 2 h at room temperature, and then washed three times with washing buffer. Antibody (100 μL/well) was added, and the plate was sealed and incubated for 2 h at room temperature and then washed three times with washing buffer. Streptavidin-HRP (100 μL/well) diluted 1:200 with diluent reagent was added to each well, and the plate was incubated in the dark for 20 min at room temperature and then washed three times using washing buffer. Substrate solution (100 μL/well) was added, and the plate was incubated in the dark for 20 min at room temperature. Finally, 50 μL of stop solution (2 N H_2_SO_4_) was added to each well, and the optical density was immediately measured using a microplate reader set to 450 nm.

### 4.7. Statistical Analysis

SPSS 13.0 (SPSS Inc., Chicago, IL, USA) was used for all Bio-plex system analyses of protein expression levels in individual samples. Associations between differences in the urine concentration of candidate proteins and different clinical parameters were analyzed using the nonparametric Mann–Whitney U test. Area under the curve (AUROC) analyses were applied to detect the optimal cutoff point that yielded the highest total accuracy in discriminating among the clinical classifications. A *p*-value less than 0.05 was considered statistically significant.

### 4.8. Network Analysis

We performed a network analysis on the distinctively expressed proteins found in each clinical group using MetaCore analytical suite version 6.13 (GeneGo, Inc., St. Joseph, MI, USA). The whole study process is briefly summarized in Figure 6. Using MetaCore’s built-in algorithms, we constructed theoretical networks, merging proteins from our experiments with those in the MetaCore database. The significance of these networks was determined by *p*-values, indicating the likelihood that a specified number of proteins from our list would correspond to certain gene nodes within the network. Pathway maps of note were then highlighted based on their significant statistical values (*p* < 0.001).

## 5. Conclusions

In summary, untargeted proteomics provides a powerful approach for detecting and quantifying metabolic shifts in patients with ADHF. Although a medium-sized sample set was analyzed herein, this is the first study to utilize a global approach to address this issue. Our results showed that EPCR was expressed at a lower level in a non-recovery group than in an early recovery group. EPCR maintains vascular barrier integrity by activating protease-activated receptor-1 signaling in the endothelium and might protect against vessel damage, such that decreased EPCR could result in non-recovery of kidney function. Future studies to explore the relationship between this protein and clinical disease are warranted.

## Figures and Tables

**Figure 1 ijms-25-02783-f001:**
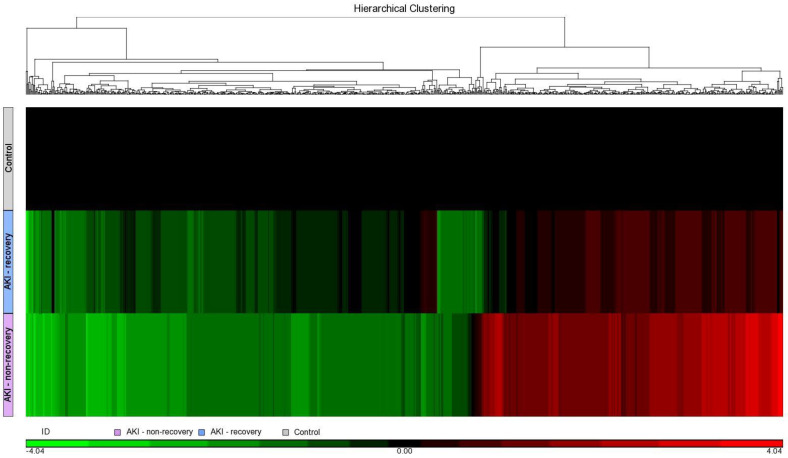
Heat map and dendrogram according to AKI non-recovery vs. AKI recovery vs. control.

**Figure 2 ijms-25-02783-f002:**
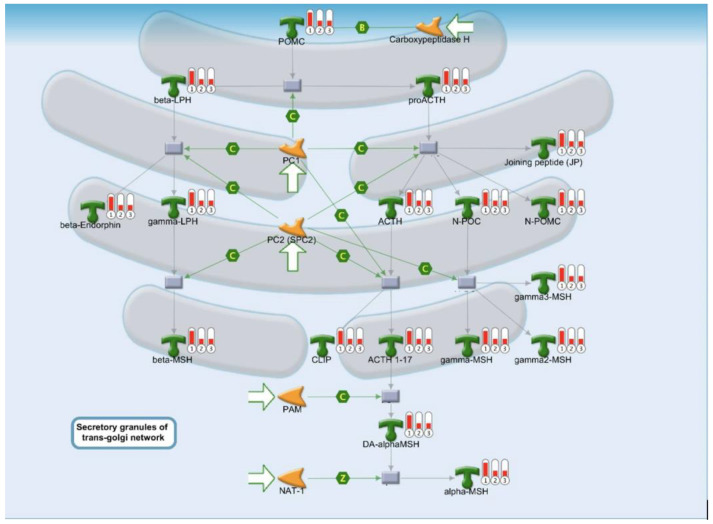
Protein folding and maturation, POMC processing.

**Figure 3 ijms-25-02783-f003:**
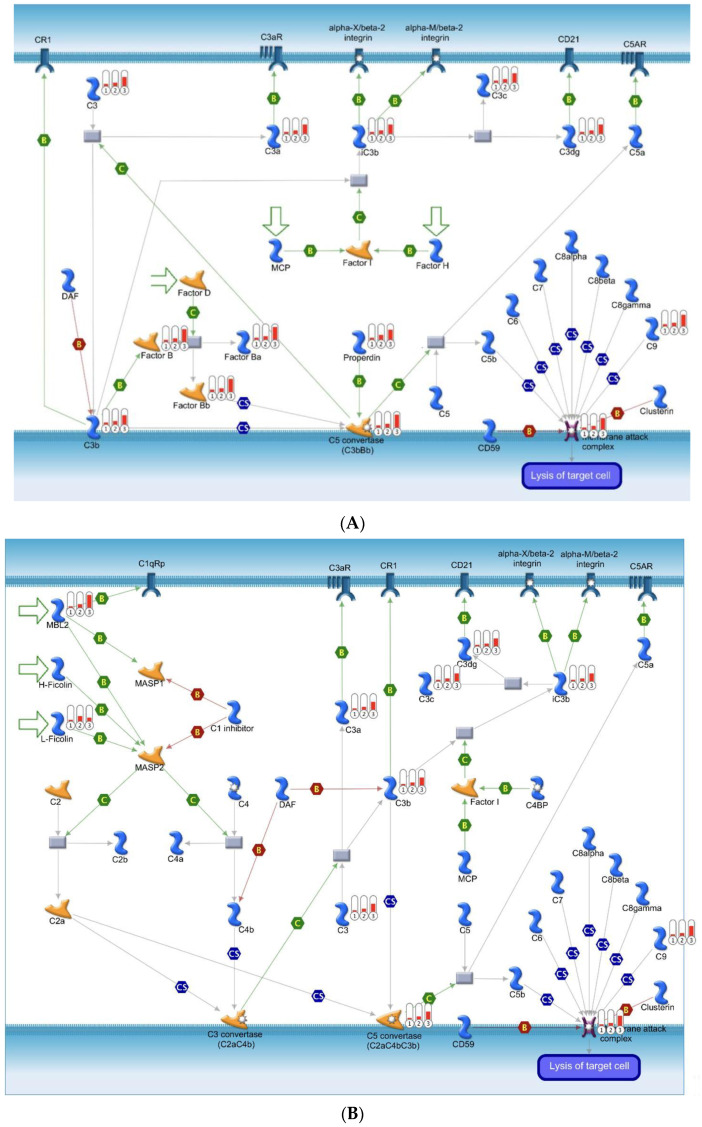
(**A**) Alternative complement pathway; (**B**) lectin-induced complement pathway.

**Figure 4 ijms-25-02783-f004:**
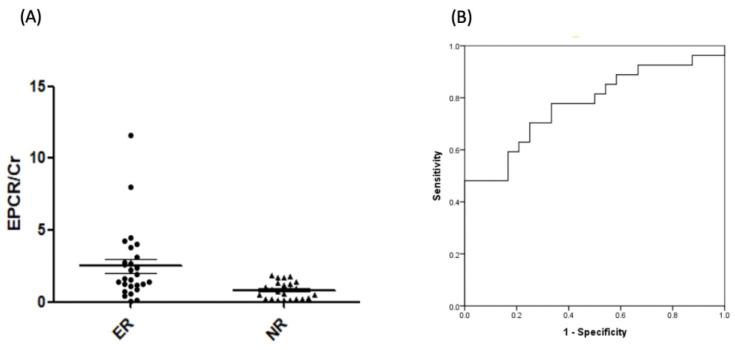
(**A**) Different expression levels of urinary EPCR according to recovery and non-recovery of AKI. (**B**) Area under the receiver operating characteristics of EPCR/Cr: 0.776 ± 0.065; 95%CI: 0.648–0.905, *p* < 0.001.

**Figure 5 ijms-25-02783-f005:**
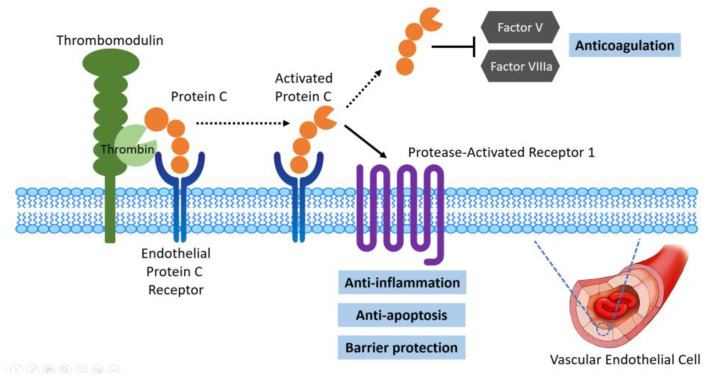
Role of endothelial protein C receptor (EPCR) in acute kidney injury. In severe acute kidney injury, endothelial damage activates the complement system and promotes thrombosis. Thrombomodulin and thrombin complex active protein C (APC) on EPCR facilitate anticoagulation. APC and EPCR cleave PAR1 to signal inflammatory and anti-apoptotic activities with barrier-protective effects. This might demonstrate the cytoprotective effects of EPCR in AKI.

**Figure 6 ijms-25-02783-f006:**
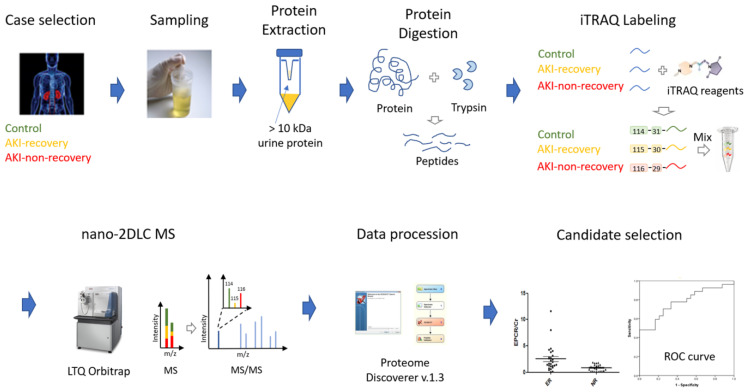
Illustration and workflow of this study.

**Table 1 ijms-25-02783-t001:** (**A**) The experiment design and urine samples used for proteomic experiment for discovery of biomarker candidates. (**B**) Protein identification and quantification results obtained from the iTRAQ experiment.

**(A)**				
**iTRAQ Tags**	**Diagnosis**	**Patient Size**	**Average Age**	**iTRAQ Labeling Tag**
114	Control	15	73.4 ± 7.4	114
115	AKI with recovery	9	74.4 ± 8.5	115
116	AKI with Non-recovery	6	71.8 ± 3.8	116
**(B)**				
		**With 1 Unique Peptide**		
	With Quantitative Results	2384	2308	
	Without Quantitative Results		76	
Protein (Peptide) of False Discovery Rate		0.01		
Both ratios of 115/114 and 116/114 tags >2		189		
**AKI > control**		589		
Both ratios of 115/114 and 116/114 tags >2		189		
Ether one ratio of 115/114 or 116/114 tags >2		400		
**AKI < control**		**537**		
Both ratios of 115/114 and 116/114 tags >2		109		
Ether one ratio of 115/114 or 116/114 tags <2		428		
No significant changes in concentrations		177		
Loss of identification in one ratio; and not consistence in two ratios		5		

**Table 2 ijms-25-02783-t002:** Demographic data and clinical characteristic based on non-recovery and recovery group patients used for verification by ELISA. *p* < 0.05 means significant.

	AKI Patients		
Variable	Non-Recovery (*n* = 24)	Recovery (*n* = 27)	*p*
* **Baseline** *			
Age (years)	67.1 ± 13.8	70.1 ± 14.0	0.442
Gender, Male (%)	17 (70.8)	20 (74.1)	1.000
Diabetes mellitus, n (%)	16 (66.7)	10 (37.0)	0.051
Serum creatinine baseline (mg/dL)	1.91 ± 0.97	1.72 ± 0.83	0.455
Serum creatinine sample day (mg/dL)	3.32 ± 1.52	3.00 ± 1.11	0.388
APACHE III score	51.7 ± 22.9	63.7 ± 33.8	0.150
SOFA score	5.1 ± 3.3	6.2 ± 4.0	0.295
Shock, n (%)	5 (20.8)	6 (22.2)	1.000
Functional class	0/20/4	4/19/4	0.145
EPCR/Cr (ng/dL)	0.82 ± 0.59	2.49 ± 2.49	0.002
* **Outcomes** *			
RRT, n (%)	7 (29.2)	0 (0.0)	0.003
LOS days	22.2 ± 21.7	14.4 ± 12.3	0.122
Mortality, n (%)	4 (16.7)	5 (18.5)	1.000

## Data Availability

Data is contained within the article and Appendix A.

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
