# Peer review of "Identification of Endothelial Cell Protein C Receptor by Urinary Proteomics as Novel Prognostic Marker in Non-Recovery Kidney Injury"

_ijms, 2024, doi:10.3390/ijms25052783_

Round 1

Reviewer 1 Report

Comments and Suggestions for Authors

This is important paper however it is lacking in some details in the results which can offer clarity to the findings 

The investigators posed the question/problem that urine biomarkers as a prognostic

and diagnostic tool I patients with AKI due to a multitude of reasons.  Perhaps searching  biomarkers that are predictive of non-recovery is an alternative.  Employing Traq 1 proteomics they screened 15 patients admitted to the ICU , a control group w no AKI, AKI with recovery and AKI without recovery. During the discovery phase they identified We identified  227 189 increased and 109 decreased proteins in two AKI groups. In the validation phase of the study they evaluated the potential of ECPCR as a potential biomarker of non recovery in a separate group of AKI patients, demonstrating that in non recovery group there was a decrease in ECPCR.  This was a peptide that was downregulated among others in the discovery phase.  This study is of significant interest however the authors need to address several questions and corrections.

Major revisions:

1.)    Table one needs to contain more information and should parallel Table 3

2.)    In the validation phase were these patients also patients with decompensated heart failure what was their diagnosis

3.)    What was the ejection fraction or classification of the CHF patients

4. )Although a decrease ECPCR may fit in the paradigm of the inflammocoagulopathic (Immune responses via the alternative complement pathway and lectin-induced pathway )  group of proteins involved in innate immunity of patients w non recovery of AKI where is the analysis of candidate proteins and ROC’s ? for example why analyze just ECPCR in the validation group why nephronectin , caspase, thrombomodulin and other whose ratios suggest candidate peptides/proteins

(Associations between differences in the urine concentration of candidate proteins and different clinical parameters were analyzed

using the nonparametric Mann-Whitney U test. Area under the curve (AUROC) analysis

were applied to detect the optimal cutoff point that yielded the highest total accuracy in

discriminating among the clinical classifications. A P-value less than 0.05 was considered statistically significant)- where are these the statistically significant values with ROC should at least be demonstrated.

Minor revisions

Line 44, statements in lines 47-50 , and 58-60  should have a references number although this is common knowledge for those in field it may not be for other a reference would be helpful for them.  

Table 2 classification has word cancer> control is cancer a typo?

Author Response

1.)    Table one needs to contain more information and should parallel Table 3

ANS: Thanks for your suggestions and reminder. The data in Table 1A showed the data of the discovery group and should be moved to the result section. Besides, the demographics in Table 2 (Table 3 in the last version) described the other patients in 2nd part of the study in which we enrolled a whole new patient group to verify our marker. Therefore, we suggest that the content might be better for separation in different tables. Thank you.

2.)    In the validation phase were these patients also patients with decompensated heart failure what was their diagnosis

ANS: Thanks for your query. These patients also had ADHF and enrolled during admission.

3.)    What was the ejection fraction or classification of the CHF patients

ANS: Thanks for your query. There is no significant difference between recovery and non-recovery groups. We added the functional class in Table 2.

  1. ) Although a decrease ECPCR may fit in the paradigm of the inflammocoagulopathic (Immune responses via the alternative complement pathway and lectin-induced pathway ) group of proteins involved in innate immunity of patients w non-recovery of AKI, where is the analysis of candidate proteins and ROC’s ? for example why analyze just ECPCR in the validation group why nephronectin, caspase, thrombomodulin and other whose ratios suggest candidate peptides/proteins (Associations between differences in the urine concentration of candidate proteins and different clinical parameters were analyzed using the nonparametric Mann-Whitney U test. Area under the curve (AUROC) analysis were applied to detect the optimal cutoff point that yielded the highest total accuracy in discriminating among the clinical classifications. A P-value less than 0.05 was considered statistically significant)- where are these the statistically significant values with ROC should at least be demonstrated.

ANS: To find the target protein to be verified by ELISA, we tested the protein by ELISA according to the possible protein function in biology and the ratio of target protein between the AKI recovery and non-recovery groups. We found that Cadherin-1, IL-18, Protein S100 A7, LPS, Uromodulin, and TNF-family were already reported in the literature. Verifying Semenogelin-1, Semenogelin-2, and apolipoprotein-C showed no association with the AKI non-recovery. Therefore, we chose the endothelial protein C receptor as a marker to detect AKI non-recovery. We added this part in the section of 2.5 “MS data processing, database searching for protein identification, protein quantification and target protein selection.”

Minor revisions

Line 44, statements in lines 47-50 , and 58-60  should have a references number although this is common knowledge for those in field it may not be for other a reference would be helpful for them. 

ANS: Thank you for your suggestions. Although the KDIGO suggested that using serum Cr as a criterion of AKI, there are some evidences showing that Cr elevated after 48-72 hours after AKI. In the IRI model, Cr elevated after 48-72 hours after IRI injury which is slower than the NGAL, IL-18 and cystatin-C. This finding was also proved in the patients after cardiac surgery, that kidney injury was induced by the open heart surgery. we added the reference as your suggestion.

Table 2 classification has word cancer> control is cancer a typo?

ANS: Thank you for your notice. We made a typo and have corrected this error in the revised manuscript.

Reviewer 2 Report

Comments and Suggestions for Authors

The results shown in this manuscript are interesting; if this marker were validated in a multicenter study, its clinical usefulness could be valued. However, my opinion is that it needs several improvements to be published.

Abstract: it lacks relevant information, for example the type of sample used, the number of patients...

Acronyms: check the acronyms, in some cases they are defined the second time they appear in the text (SCr lines 42 and 43), in others the definition is repeated (KDIGO lines 41 and 92). These are just some examples, check the full text.

lines 44-45: better explain the limitation of creatinine, what does it mean that it rises 48-72 hours after kidney damage?

This manuscript does not include an objective

Table 1: the n is too small to be able to draw conclusions, although it is later expanded to confirm these results. In this table, I believe that the characteristics of the patients are missing, since the differences could be due solely to differences in sex or differences in a pathology that exists more in one group than in another (diabetes mellitus?).

Table 2: I cannot find the information of this table in the text. Why is cancer mentioned if it has been said that it was an exclusion criterion?

3.2: I do not understand this as a paragraph, rather it would be the introduction to two subsections 3.2.1 protein folding and .3.2.2. inflammation. I think that if so much reference is going to be made to these results, they should somehow go in a figure (a summary table or a diagram) since the text, by itself, is complicated to follow.

Table 3: where are the ELISA results? If it is the main result, it would be necessary to know the values, this would help other researchers to check if the EPCR levels in their patients resemble recovery or non-recovery. Furthermore, this table shows a percentage of patients with diabetes mellitus very close to a statistically significant difference. Have you verified that the lower excretion is not due to some effect related to this pathology?

I think the discussion focuses on very general terms that can overlap with the introduction rather than explaining the possible usefulness of the analyzed biomarker, especially in points 4.1 and 4.2.

Figure 1 of the supplementary material: why is NGAL mentioned if what appears in the figure is EPCR/Cr 

Comments on the Quality of English Language

The work is written clearly and is well understood but requires a general review to improve aspects such as the issue of acronyms (explained in the comments) and that on some occasions there are extra spaces between two words.

Author Response

Abstract: it lacks relevant information, for example the type of sample used, the number of patients.

ANS: Thank you for your kind review and comment. We have revised our abstract as your suggestion.

Acronyms: check the acronyms, in some cases they are defined the second time they appear in the text (SCr lines 42 and 43), in others the definition is repeated (KDIGO lines 41 and 92). These are just some examples, check the full text.

ANS: Thanks for your notice; we changed SCr to serum Cr to avoid misunderstanding in the whole manuscript.

lines 44-45: better explain the limitation of creatinine, what does it mean that it rises 48-72 hours after kidney damage?

ANS: Thanks for your query and suggestion. Although the KDIGO suggested that using serum Cr as a criterion of AKI, there are some evidence showing that Cr elevated after 48-72 hours after AKI. In the IRI model, Cr elevated after 48-72 hours after IRI injury which is slower than the NGAL, IL-18 and cystatin-C. This finding was also proved in the patients after cardiac surgery, that kidney injury was induced by the open heart surgery. (reference: 20216399 Anesthesiology April 2010, Vol. 112, 998–1004.)

This manuscript does not include an objective

ANS: Thanks for your suggestions, we aim to find a novel AKI biomarker in patients with ADHF. Since the known biomarkers cannot predict non-recovery AKI accurately, a novel biomarker would be benefit in the clinical practice for ADHF patients. We emphasized that in the section 1.1.

Table 1: the n is too small to be able to draw conclusions, although it is later expanded to confirm these results. In this table, I believe that the characteristics of the patients are missing, since the differences could be due solely to differences in sex or differences in a pathology that exists more in one group than in another (diabetes mellitus?).

ANS: Thanks for your suggestions and reminder. The data in Table 1A showed the data of the discovery group and should be moved to the result section. In the discovery group, we selected the patients who presented with clinical AKI apparently and divided them into recovery and non-recovery. Besides, the demographics in Table 2 (Table 3 in the first version) described the other patients who were different from the discovery group. We also issued the small case number in our limitation. Thank you.

Table 2: I cannot find the information of this table in the text. Why is cancer mentioned if it has been said that it was an exclusion criterion?

ANS: Thank you for your notice. We made a typo and have corrected this error in the revised manuscript.

3.2: I do not understand this as a paragraph, rather it would be the introduction to two subsections 3.2.1 protein folding and .3.2.2. inflammation. I think that if so much reference is going to be made to these results, they should somehow go in a figure (a summary table or a diagram) since the text, by itself, is complicated to follow.

ANS: Thank you for your comment. We have made some modifications for better understanding and to enrich the article's content. First, we move it from the supplement to the main manuscript for better reading. Second, we rewrite the sentence to emphasize the effects of these pathways in AKI. The protein folding pathway was affected between AKI and non-AKI without significant changes between recovery and non-recovery. However, the complex pathways were expressed differently between the recovery and non-recovery, which might hint that the complex system involved the mechanism of cell death, which retard the kidney recovery.

Table 3: where are the ELISA results? If it is the main result, it would be necessary to know the values, this would help other researchers to check if the EPCR levels in their patients resemble recovery or non-recovery. Furthermore, this table shows a percentage of patients with diabetes mellitus very close to a statistically significant difference. Have you verified that the lower excretion is not due to some effect related to this pathology?

ANS: Thanks for your notice. To make it better readable and for the researcher to know the EPCR/Cr level, we moved the result of ELISA from the supplementary to the result section (Now as Figure 5). In the discovery phase, it’s better to provide the ratio of EPCR to Cr, especially when using the urine sample. Compared to serum biomarkers, the protein level in urine should be corrected with urine Cr, especially in the clinical samples. These urine concentrations might be affected by Lasix, IV fluid, or water restriction. And the level of EPCR/Cr is 2.03±0.51 1 vs. 38±0.25 (P=0.037) in non-DM vs. DM accordingly.

I think the discussion focuses on very general terms that can overlap with the introduction rather than explaining the possible usefulness of the analyzed biomarker, especially in points 4.1 and 4.2.

ANS: Thank you for your comment. In the section of 4.1 and 4.2, we described the advances and problems while using iTRAQ to discovery the biomarkers in the urine compared to traditional method such as 2D gel. We suggested this is an important information for other researchers to repeat this method further experiment.

Figure 1 of the supplementary material: why is NGAL mentioned if what appears in the figure is EPCR/Cr 

ANS: Thanks for your careful inspection. We revised the legend of Figure 5, which was originally in supplement material, and deleted the NGAL.

Reviewer 3 Report

Comments and Suggestions for Authors

In this study, the authors presented endothelial cell protein C receptor by urinary proteomics as a novel prognostic marker in non-recovery kidney injury.

This is innovative study with promising results, but I have some comments and suggestions.

My main concern is the number of patients that were included in the study. It is only 6 and 9  per experimental group. They were enrolled from Dec 2016 to Jan 2020. Such small samples can seriously question the validity of results obtained. How do you comment on this sample size?

Table 2 looks pretty confusing. I didnʼt understand what did you mean with ...cancer˃normal (I)... and ...cancer<normal (II...) in classification. Patients with cancer were excluded from the study.

Also Table 1B should be moved to the part of the text that is related to.

According to your analysis such large number of proteins, did you recognize EPCR as the only one with the potential as a novel marker in non-recovery AKI, or it was the only one you tested by ELISA?

Comments on the Quality of English Language

Minor editing of English language required.

Author Response

My main concern is the number of patients that were included in the study. It is only 6 and 9  per experimental group. They were enrolled from Dec 2016 to Jan 2020. Such small samples can seriously question the validity of results obtained. How do you comment on this sample size?

ANS: Thank you for your kind review and comment. The small patient sizes were initially used for biomarker candidate discovery, and further verification and validation were performed using larger number of patient size. We also added the limitation of sample size in the section of restriction.

Table 2 looks pretty confusing. I didnʼt understand what did you mean with ...cancer˃normal (I)... and ...cancer<normal (II...) in classification. Patients with cancer were excluded from the study.

ANS: Thank you for your notice. We made a typo and have corrected this error in the revised manuscript.

Also Table 1B should be moved to the part of the text that is related to.

ANS: Thank you for your kind notice. We moved it to the result section.

According to your analysis such large number of proteins, did you recognize EPCR as the only one with the potential as a novel marker in non-recovery AKI, or it was the only one you tested by ELISA?

ANS: Thank you for your query. To find the target protein to be verified by ELISA, we tested the protein by ELISA according to the possible protein function in biology and the ratio of target protein between the AKI recovery and non-recovery groups. We found that Cadherin-1, IL-18, Protein S100 A7, LPS, Uromodulin, and TNF-family were already reported in the literature. Verifying Semenogelin-1, Semenogelin-2, and apolipoprotein-C showed no association with the AKI non-recovery. Therefore, we chose the endothelial protein C receptor as a marker to detect AKI non-recovery. We added this part in the section of 2.5 “MS data processing, database searching for protein identification, protein quantification and target protein selection.

Round 2

Reviewer 1 Report

Comments and Suggestions for Authors

The answers to the questions and revisions have improved paper and manuscript is acceptable to publish

Author Response

Thanks for your review and we appreciate your help.

Reviewer 2 Report

Comments and Suggestions for Authors

The authors have made most of the proposed changes, although they still have some small errors that should be corrected.

The aspect that worries me the most is that the objective of the work is mixed with the explanation of the introduction (in point 1.1.). Although this is not a problem that prevents its publication, I think it is something that could be improved or highlighted the objective in some way.

I would recommend reviewing the text in detail to find possible errors, I have detected some:

- The issue of the KDIGO acronym has not been corrected, the definition appears on line 41, so in line 97 only the acronym should be used.

- Line 85: 2. the dot is in red, change to black

- there are two figures 2. Change the one on line 445 to figure 6

Author Response

Thanks for your careful inspection. We revised the manuscript per your suggestion and highlighted our aims again in section 2.3, just before the method. We appreciate your help and review.

Reviewer 3 Report

Comments and Suggestions for Authors

The authors responded to all comments.

Comments on the Quality of English Language

Minor editing of English language required.

Author Response

ANS: Thanks for your review and we appreciate your help. We have sent the manuscript for editing English by professor English editing service before submission and have rechecked the revised manuscript again. We also attached the invoice in the supplementary file (not for publication) for your information. The English quality was not mentioned by the other two reviewers. I hope this explanation can be accepted by the reviewer. Thank you for your review.